# Demonstration of Tunable Control over a Delayed-Release Vaccine Using Atomic Layer Deposition

**DOI:** 10.3390/vaccines12070761

**Published:** 2024-07-11

**Authors:** Sky W. Brubaker, Isabella R. Walters, Emily M. Hite, Lorena R. Antunez, Emma L. Palm, Hans H. Funke, Bryan L. Steadman

**Affiliations:** VitriVax, Inc., 5435 Airport Blvd Suite 106, Boulder, CO 80301, USA; iwalters@vitrivaxbio.com (I.R.W.); ehite@vitrivaxbio.com (E.M.H.); lnapolitano@vitrivaxbio.com (L.R.A.); epalm@vitrivaxbio.com (E.L.P.); hfunke@vitrivaxbio.com (H.H.F.); bsteadman@vitrivaxbio.com (B.L.S.)

**Keywords:** single-administration vaccines, atomic layer deposition, controlled antigen release, in vitro–in vivo correlation

## Abstract

Many vaccines require multiple doses for full efficacy, posing a barrier for patient adherence and protection. One solution to achieve full vaccination may be attained with single-administration vaccines containing multiple controlled release doses. In this study, delayed-release vaccines were generated using atomic layer deposition (ALD) to coat antigen-containing powders with alumina. Using in vitro and in vivo methods, we show that increasing the coat thickness controls the kinetics of antigen release and antibody response, ranging from weeks to months. Our results establish an in vitro–in vivo correlation with a level of tunable control over the antigen release and antibody response times with the potential to impact future vaccine design.

## 1. Introduction

There is great promise in developing new vaccine technologies with improved safety, efficacy, and durability. Current vaccines often require multiple doses spaced over time to provide optimal protection [1,2]. Poor compliance with vaccine schedules can reduce efficacy, posing a critical public health challenge from preventable infectious diseases throughout the world [3]. To overcome this hurdle, controlled-release technology is a promising strategy for delivering more than one vaccine dose on a predictable schedule using a single administration [4]. However, precision in tuning the controlled-release technologies to dictate the kinetics of the vaccine release and immune response times remains a challenge.

Several controlled-release vaccine technologies have emerged [5,6,7], which differ in their release profiles. Sustained release profiles provide continuous delivery of the antigen over a long period of time [8]. Pulsatile release profiles are characterized by a delay period and one or more pulses of antigen. Combining vaccine materials with different delay periods in a single dose may achieve an antigen delivery profile that mimics exposure from temporally spaced, repeated bolus injections (e.g., a standard vaccine schedule), thus approaching 100% efficacy without the need for multiple vaccine administrations.

Spray drying and ALD have previously been combined to achieve thermostability and delayed release in vivo with a single administration of a human papillomavirus (HPV) vaccine [9,10]. The vaccine preparations of HPV L1 capsomers were spray dried to make thermostable powders that were subsequently coated with alumina by ALD. Spray-dried particles have previously been imaged by scanning electron microscopy before and after ALD coating to establish particle size and shape characteristics [9]. In addition, transition electron microscopy was used to confirm the estimates that sequential ALD reactions deposit roughly 2.3-Å-thick layers of alumina on the particle surface [9]. The alumina coating prolonged the antigen retention at the site of injection, suggesting that the antigen release was delayed in vivo [9]. Furthermore, animals vaccinated using ALD-coated powders elicited neutralizing antibody titers that were comparable to the animals receiving a conventional prime and boost administration of the vaccine [9,10]. However, the range of achievable immune response times remains to be determined for ALD-coated powders.

## 2. Materials and Methods

### 2.1. Spray Drying, ALD Coating, and Other Materials

Intermediate spray-dried powders were made using a Buchi Mini B-290 spray dryer with a B-296 Dehumidifier (BUCHI, New Castle, DE, USA). The proprietary formulations contained 13% total solids, including 1% ovalbumin (OVA) (InvivoGen, San Diego, CA, USA, #vac-pova-100). Spray-drying setpoints were chosen to target the desired powder properties, including particle size, residual moisture content, and spray dryer yield. ALD-coated powders were generated using the intermediate spray-dried powders in a custom, mechanically agitated fluidized-bed ALD reactor, using trimethylaluminum and water as precursors [9,10]. Purge steps were employed after each precursor exposure to avoid chemical vapor deposition. The coating was performed at 50 °C, using argon as the fluidization gas. The number of ALD cycles used to coat powders in this study were 50, 100, 250, 500, and 1000.

The Alhydrogel^®^ (InvivoGen, San Diego, CA, USA, #vac-alu-50) and OVA used in the animal studies were prepared following the manufacturer’s instruction. Briefly, Alhydrogel^®^ adjuvant 2% was combined with the OVA antigen in a 1:1 volume, mixed, and allowed to adsorb for 30 min at room temperature prior to administration. For the animal groups receiving Alhydrogel^®^, the mass of aluminum administered was approximately 11.6 µg per 500 ng dose of OVA (Figure 1C,D) or 1.6 µg per 250 ng dose of OVA (Figure 1E–G).

### 2.2. Vaccine Administration and Animal Studies

All animal studies were conducted under approval from the Institutional Animal Care and Use Committee at the University of Colorado, Boulder, CO (protocol #2835). Vaccine administrations were given by intramuscular injection to the right flank of female C57BL/6J mice aged 6–8 weeks (The Jackson Laboratory, Bar Harbor, ME, USA, Strain #000664). The dosing was based on OVA mass and delivered with diluent (saline containing 6% hydroxyethyl starch) in a final volume of 50 µL. The serum was isolated from submandibular blood samples using gel separation tubes (Sarstedt, Nümbrecht, Germany, #41.1378.005) and stored at −80 °C prior to analysis.

### 2.3. Quantitation of Antibody Titers by ELISA

The immunized mouse serum samples were analyzed by an indirect, enzyme-linked immunosorbent assay (ELISA) to determine anti-OVA total IgG, IgG1, IgG2c, IgA, and IgM antibody titers. Nunc 96-well flat bottom high-binding plates (Thermo Scientific, Norristown, PA, USA, #442404) were coated with 5 µg/well of OVA (Fisher, Hampton, NH, USA, #BP2535-5) in phosphate-buffered saline (PBS), incubated at 4 °C overnight, and then rinsed four times using a wash buffer (0.05% Tween 20 in PBS). The plates were then incubated with an assay-blocking buffer (3% BSA, 0.05% Tween 20 in PBS) at room temperature for two hours. A standard curve was established with a dilution series of calibrated primary control (total IgG: Biolegend, San Diego, CA, USA, #520501, IgG1: Chondrex, Woodinville, WA, USA, #7093, IgG2c: Chondrex, #7109, and IgM: Chondrex, #7097). The calibrated primary control and mouse sera samples were incubated for 1 hour at room temperature, followed by a wash step to remove the unbound antibody. Horseradish peroxide-conjugated secondary antibody (Biolegend, #405306 @ 1:2000 dilution; Southern Biotech, Birmingham, AL, USA, #1071-05 @ 1:4000 dilution; Southern Biotech, #1078-05 @ 1:4000 dilution; and Southern Biotech, #1021-05 @ 1:1000 dilution) was added (50 µL/well) and incubated for one hour at room temperature. Excess secondary antibody was washed off, and the plates were developed for 20 min using Ultra TMB (Thermo Fisher, #34028) and quenched using sulfuric acid (H_2_SO_4_), per the manufacturer’s protocol. Absorbance was measured at 450 nm and 650 nm on a BioTek Synergy plate reader (Agilent, Santa Clara, CA, USA) five minutes after the addition of H_2_SO_4_. The standard curves and interpolated data for each plate were independently generated using Gen5 software (Agilent, Santa Clara, CA, USA).

### 2.4. Accelerated Dissolution of ALD Coated Particles In Vitro

ALD-coated particles were suspended at a concentration of 10 mg/mL buffered solution in Lo-bind microcentrifuge tubes (Eppendorf, Hamburg, Germany, #022431102) and agitated on an Eppendorf Thermomixer C set at 50 °C and 950 rpm. The dissolutions were performed at 50 °C to fully capture the dissolution profiles of the ALD-coated particles in a reasonable time frame while minimizing the risk of evaporation and microbial contamination. The supernatants were sampled at the indicated times after brief centrifuging to pellet the ALD powder. The volume removed was replaced with an equivalent volume of fresh buffer, the pellets were resuspended, and the tubes were placed back in the Thermomixer. The supernatant samples were added to a 96-well assay plate, and content released from the ALD-coated particles was measured using a Fluoraldehyde o-phthaldialdehyde (OPA) Assay Reagent Mix (Thermo Scientific, #26025) per the manufacturer’s instructions. Fluorescence was measured after excitation at 360 nm and detected at 450 nm using a BioTek Synergy plate reader (Agilent, Santa Clara, CA, USA). Data are presented as percent recovery and calculated by normalizing the fluorescent signal against the maximum signal from each sample. The data were analyzed using four-parameter logistic (4PL) regressions.

### 2.5. Statistical Analysis

The Prism (Graph Pad, Boston, MA, USA) software version 10.2.3 was used for the statistical analysis and for the linear as well as nonlinear regression analyses. The nonparametric statistical analysis was conducted using a Kruskal-Wallis ANOVA with Dunn’s post-hoc test α = 0.05 [11].

## 3. Results

The tunable release of antigen from ALD-coated powders was shown using an accelerated in vitro dissolution assay. The coated powders were incubated in a dissolution buffer at 50 °C and sampled over time to monitor the release. Powders coated with 100 cycles of ALD demonstrate a delay phase followed by a release phase (Figure 1A). Nonlinear regression (4PL) analysis of the data generated a curve and parameters reporting on the initial and final percent release, the time to the inflection point, and the slope of the curve at the inflection point (Appendix A). The initial release is indicative of the coat quality or the containment of the antigen after the suspension of the powder. The inflection point represents the length of time it takes to release roughly 50 percent of the total content.

To achieve products with different release times, the number of ALD cycles used for alumina coating can be adjusted to increase or decrease the coat thickness [9]. The in vitro dissolution assay confirmed that the timing of release is controlled by the coat thickness, using coated powders generated with 50–1000 ALD cycles (Figure 1B). The dissolution profiles showed that all the powders had a robust coating (greater than 90% initial containment) and that increasing the number of ALD cycles resulted in increasingly delayed releases (Appendix A).

To explore the potential for an early immune response from ALD-coated vaccine powders, mice were either administered coated powder that released quickly in vitro (50 ALD cycles; Figure 1B) or uncoated powder. The powders were spray dried in a buffer formulation containing the model antigen ovalbumin (OVA) but without adjuvant. The coated and uncoated powders were then compared against a liquid formulation of OVA and Alhydrogel^®^, a conventional aluminum-based adjuvant. The total IgG antibody responses to OVA were similar between the animals receiving 50-cycle ALD powders and Alhydrogel^®^ (Figure 1C). Lower IgG titers were detected in the animals treated with uncoated spray-dried powders, likely due to the absence of the adjuvant or aluminum coating. There was a statistically significant difference in the total IgG titer at two weeks between the animals receiving powders that were coated vs. the animals that received the uncoated powders (**, *p* 0.0044); however, there was no significant difference when comparing 50-cycle ALD against the Alhydrogel^®^-formulated OVA (ns, *p* 0.6093 Figure 1D). A subset of antibody isotypes was measured, and the increase in the total IgG can be predominantly attributed to the increased production of OVA-specific IgG1 (Figure 1E–G). These data confirm that ALD-coated vaccines effectively trigger a robust humoral response with the antibody class switching predominantly toward IgG1 [12]. Furthermore, the data establish that 50-cycle powder induces an early antibody response that mimics a conventionally adjuvanted liquid vaccine formulation (i.e., the kinetic comparison with the Alhydrogel^®^-formulated OVA in these experiments is similar to the kinetics of ALD 50-cycle powder). In summary, ALD-coated vaccine powders are immunogenic and can achieve similar early response kinetics comparable to a conventionally adjuvanted liquid vaccine formulation.

A challenge in expanding the use of ALD for single-administration vaccines is demonstrating its ability to control payload release for predictable delayed immune responses in vivo. Similar to the delayed in vitro vaccine release with increasing coat thickness (Figure 1B), the in vivo immune response kinetics for the same coated powders confirmed increasingly delayed immune response with increasing coat thickness (Figure 2). The geometric mean titers (GMT) of OVA-specific IgG1 increased rapidly following the administration of 50-cycle ALD powders, whereas the increase in titer was delayed following the administration of 250-, 500-, or 1000-cycle ALD powders (Figure 2A).

There was not a statistically significant difference in the peak antibody titer responses compared between 50 and 250 (ns, *p* > 0.9999)- or 50 and 500 (ns, *p* > 0.9999)-cycle ALD powders (Figure 2B). Thus, similar immunogenicity can be elicited from the delayed-release ALD powders as compared to the early response elicited from 50-cycle powder. In contrast, the median peak titer was significantly lower among the animals that received 1000-cycle ALD powders as compared to the animals that received 50-cycle powders (**, *p* 0.0024). Two of the animals in this group responded robustly, whereas the rest were classified as non-responders (peak titers < 10^3^ ng/mL). It is possible that the experiment was terminated before the vaccine release occurred in the non-responder mice, or the results may indicate that higher doses are required for powders coated with many ALD cycles. The median time to peak titer (using the cutoff > 10^3^ ng/mL to exclude non-responders) for 50-, 250-, 500-, and 1000-cycle powders was 4, 15, 30, and 38 weeks, respectively (Figure 2C). This result demonstrates that increasing the number of ALD cycles delays the antibody response in vivo. However, the peak antibody titer in each animal may not reflect the onset of antigen-specific antibody production.

Seroconversion, a detectable increase in titer over pre-vaccination levels, is a better measure for the onset of the antibody response. Thus, seroconversion was deemed a more suitable parameter for establishing an in vitro–in vivo correlation between the time to the antigen release and the time to the antibody response. The threshold determination for seroconversion used in this study was a two-log increase in OVA-specific IgG1 titer. Increasing the number of ALD cycles slowed the rate of seroconversion in vivo (Figure 2D) with median times to seroconversion for 50-, 250-, 500-, and 1000-cycle powders of 2, 4, 13, and 36 weeks, respectively (Figure 2E). The in vitro release kinetics (Figure 1B; Appendix A) for different coating thicknesses linearly correlates with the time to in vivo seroconversion (Figure 2F) as shown in Figure 2E (R^2^ = 0.9031). Thus, vaccines which induce tunable, predictably delayed immune responses can be formulated by encapsulating spray-dried powders containing antigen with ALD alumina of varying thicknesses.

## 4. Discussion

The use of atomic layer deposition as a method to coat a thermostable dried vaccine product is a relatively new. Prior research has demonstrated how the approach maintains thermostability for three clinically relevant antigens, the L1 capsomere proteins from HPV 16, 18, and 31 [10]. In addition, ALD coating has been shown to impart delayed-release capabilities with the potential for creating single-administration vaccines [9].

In this report, we expand on these previous studies to show the temporal control of vaccine-induced immune response using ALD alumina-coated powders. An accelerated in vitro dissolution method was developed to characterize coated powders and was used to demonstrate that increasing ALD cycle numbers delayed the vaccine release (Figure 1A,B). Furthermore, the time to release in vitro was shown to correlate with the antibody response times in vivo (Figure 2F).

Robust and delayed antibody response times between 2 and 13 weeks were achieved by varying the thickness of the ALD coating. In an extreme example using particles coated with 1000 cycles, the data showed that response times 30 weeks after injection may be achievable with ALD technology (Figure 2). While impressive, this may not be necessary or desirable when considering the design of a clinical vaccine product. However, combining ALD-coated vaccines in a single administration, providing two doses (an immediate and delayed dose) separated by eight weeks, may be more desirable. The differently coated products could deliver the same antigen for a classic prime and boost, or they could contain different antigens depending on the goal. Future combination studies testing these concepts are needed to uncover factors to consider when making a combined product.

The performance characteristics of ALD-coated vaccines compared against competing vaccine platforms or adjuvants is another area for future research. Many head-to-head comparisons between technologies using different types of antigens are likely needed. Alhydrogel^®^ and other aluminum-based adjuvants are generally considered to direct strong humoral or Th2 responses. It will be interesting to understand whether ALD-coated vaccines behave similarly or whether they stimulate distinct cellular immune responses.

Other approaches to achieving controlled-release vaccine technology have been reported previously [13,14,15,16,17,18,19,20,21]. To our knowledge, this is the first example demonstrating controlled-release vaccine technology that stimulates a delayed antibody response in the absence of any priming dose. This is critical for product development to obtain in vivo response times that align with a desired immunization schedule. This report also describes an approach for characterizing ALD-coated vaccine powders that may guide rational vaccine design to elicit immune responses that mimic repeated administrations of a standard vaccine product. ALD-coated vaccines provide a critical step toward generating single-administration products that can have a significant impact against preventable infectious diseases.

## 5. Conclusions

This study provides an example of how to establish an in vitro to in vivo correlation for the characterization of delayed-release vaccine technologies. Specifically, we show that increasing the thickness of ALD-coated vaccines results in longer delays before release in vitro that correlate with longer delays in antibody response times in vivo. The results demonstrate that ALD coating technology can be used to achieve tunable control over a wide range of immune response times from a vaccine.

## Figures and Tables

**Figure 1 vaccines-12-00761-f001:**
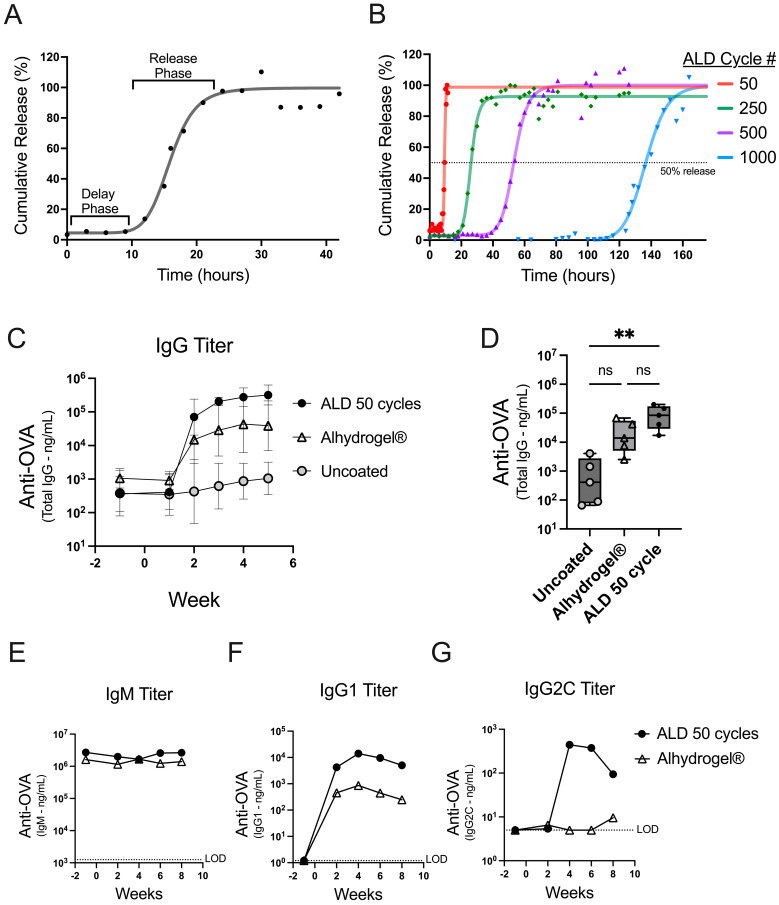
In vitro dissolution and early antibody responses kinetics. (**A**) The in vitro dissolution from powder coated with 100 ALD cycles is presented as a percentage of the total content released over time. Individual data points are depicted (dots) with a four-parameter, nonlinear logistic (4PL) regression (line). (**B**) The in vitro dissolution curves are shown for powders coated with 50, 250, 500, and 1000 ALD cycles. A dotted line at 50% gives an approximation of the inflection point for each 4PL regression (Appendix A). (**C**,**D**) Mice (*n* = 5) were vaccinated on day 0 with 500 ng OVA either from uncoated spray-dried powder (gray circles) or a 50-cycle ALD-coated powder (black circles), or with an Alhydrogel^®^ formulation (open triangles). (**C**) OVA-specific total IgG from serum was plotted over time as the GMT for each group with 95% confidence intervals. (**D**) OVA-specific total IgG serum titers at two weeks post-injection are shown as box plots (+/−; min/max), with an overlay of individual data points. (**E**–**G**) Mice (*n* = 5) were vaccinated on day 0 with 250 ng OVA either from 50-cycle ALD-coated powder (black circles) or formulated with Alhydrogel^®^ (open triangles). The serum from the individual mice was pooled at each timepoint, and titer concentrations for each timepoint are reported for OVA-specific IgM (**E**), IgG1 (**F**), or IgG2C (**G**) antibodies. The statistical analysis for panel D was performed by Kruskal-Wallis ANOVA with Dunn’s post-hoc test (a = 0.05) ** indicates *p*-value = 0.0044, ns indicates not significant, and # indicates number.

**Figure 2 vaccines-12-00761-f002:**
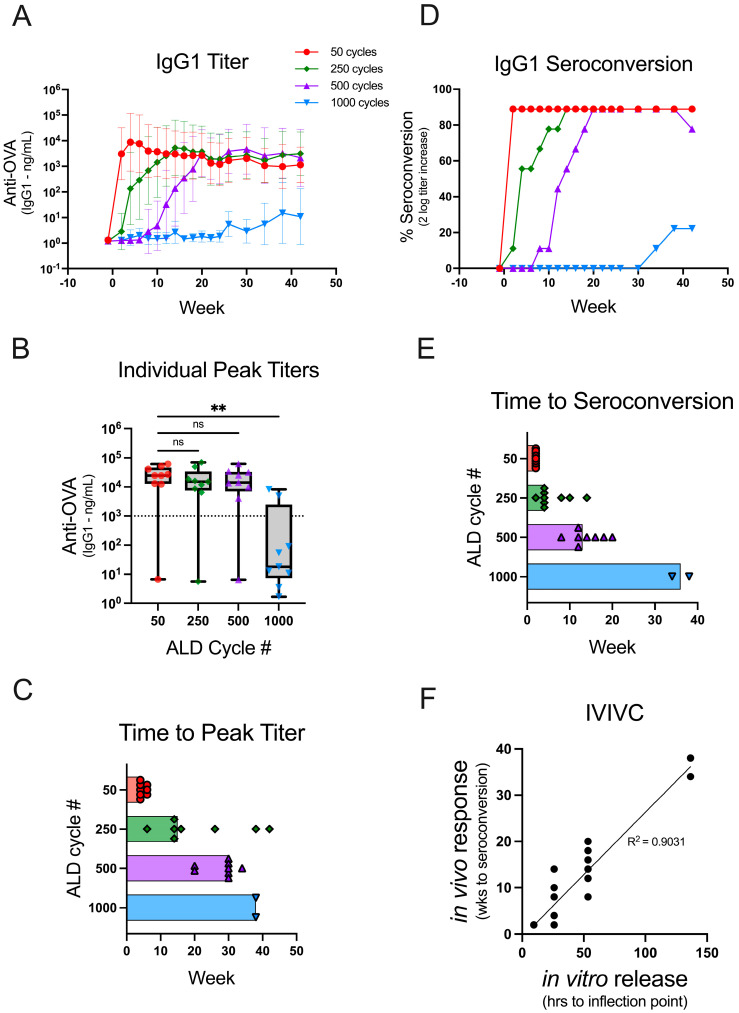
Coat thickness delays the antibody response times in vivo. Mice (*n* = 9) were administered a single 250 ng dose of OVA on day 0 using reconstituted powders coated with 50 (red circle), 250 (green diamond), 500 (purple triangle), or 1000 (blue triangle) cycles of ALD, which increases the coat thickness. Individual animal sera were analyzed for OVA-specific IgG1 antibody titers over the course of 42 weeks. (**A**) The GMT for each animal group is depicted over time with 95% confidence intervals. (**B**) The peak titers are shown for each group as box plots (+/−; min/max) with an overlay of individual data points. The dotted line on the graph at 10^3^ ng/mL represents a cutoff between the responders and non-responders. (**C**) The time to peak titer for each individual mouse (excluding the non-responders) is shown, and the bars extend to the median time to peak titer for the group. (**D**) The percent seroconversion (as determined by a two-log increase in titer over pre-vaccination) is shown for each group over the course of 42 weeks. (**E**) The time to seroconversion for each individual mouse is shown, and the bars extend to the median time to seroconversion for the group. (**F**) An in vitro to in vivo correlation was drawn by plotting the time to the inflection point of the in vitro dissolution (Figure 1B, Appendix A) against the individual mouse times to seroconversion. The correlation line was generated using a simple linear regression. Statistical analysis for panel B was performed by Kruskal-Wallis ANOVA with Dunn’s post-hoc test (α = 0.05) ** indicates *p*-value = 0.0024, ns indicates not significant, and # indicates number.

## Data Availability

The original contributions presented in the study are included in the article/Appendix A, further inquiries can be directed to the corresponding author.

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
