# Peer review of "Demonstration of Tunable Control over a Delayed-Release Vaccine Using Atomic Layer Deposition"

_vaccines, 2024, doi:10.3390/vaccines12070761_

Round 1

Reviewer 1 Report

Comments and Suggestions for Authors

Summary:

This paper describes an interesting approach for delaying the release of a model antigen, ovalbumin (OVA) by coating it with aluminum using atomic layer deposition. The novelty of this approach is somewhat reduced by the existence of the authors’ 2020 npj Vaccines paper that shows the technique being used with a clinically relevant antigen and provides more extensive characterization of the system and immune response. Nevertheless, this paper provides new insights that contribute value to the field. Some of the statistical analysis has to be improved and more discussion is needed to address unanswered questions. It also seems like an obvious missed opportunity to include multiple doses (e.g., ALD with different numbers of cycles) to show that they can achieve an anamnestic immune response. Please find list of major and minor issues below:

Major Issues:

1.    The use of accelerated release experiments is not well-justified and it does not seem that running release experiments at 37 C would pose any long-term challenges as the current release times under accelerated conditions are on the order of tens of hours. The authors should either run the experiments at 37 C or provide better justification for why accelerated release was the correct choice.

22. There are no characterization methods used to show what the coated particles look like. At a minimum, they should use scanning electron microscopy images or something similar to show particles before and after coating. Something like this may be included in reference #9 but this paper should be self-contained and not dependent on another paper (that uses a different antigen as well).

33.    There are many places where a single data point is reported, lacking standard deviations and/or error bars (e.g., all of Fig 1 except for 1D, Figure 2A,D, Supp Tables 1 & 2). This needs to be corrected.

44.    The authors should discuss that the favorable results obtained in the paper are due to the powder acting like nano alum, which typically outperforms Alhydrogel as an adjuvant.

55.    The evidence for antibody class switching is very weak, consisting only of IgM and IgG titers in which the IgM titers exceed IgG1 titers by more than 100-fold. It is also unclear if the alum and experimental group are statistically significantly different since there are no error bars or indications of statistical significance. Even if it were, the data should be taken in context since IgM titers are also higher for the ALD 50 group. I would recommend excluding the comment on like 155. Similarly, I do not see evidence in Figure 1 for the response beginning earlier. The ALD 50 and alum groups both appear to respond immediately. If the authors want to keep the corresponding comment on lines 156 and 157, they need to support the claim with data.

66.    The extensive discussion of seroconversion does not make sense and should be removed since there is no key level that is known to be protective from OVA since it is not infectious. Therefore, all these sub-figures and descriptions do is binarize quantitative data without adding value.

Minor Issues:

17.    The authors should comment on the suitability of this method for encapsulating vaccine antigens rather than a model protein. For example, it seems like the fabrication conditions (50 C) would/could change the conformation of the antigen and thus its ability to confer immunity.

28.    The authors should discuss what a reasonable number of cycles would be for a commercial product. For example, would 1,000 cycles be commercially viable or too expensive?

39.    The legend in Figure 1B is missing the color markings. I believe two asterisks are also missing from the comparator bar in Figures 1D and 2B.

410.    Can the authors please explain how they are getting antibodies in ng/mL? I am not sure how this number can be obtained without a number of assumptions and it is not standard to report this type of unit.

Author Response

Dear Reviewer 1,

Thank you for your thoughtful and kind assessment of our submitted manuscript. We agree with your summary statement and would like to highlight that an expanded discussion has been included regarding future studies that can be conducted to further characterize the technology. Please find the responses to your comments below. We hope that you find the responses and revisions to the manuscript to be satisfactory.

Major Issues:

  •  The use of accelerated release experiments is not well-justified and it does not seem that running release experiments at 37 C would pose any long-term challenges as the current release times under accelerated conditions are on the order of tens of hours. The authors should either run the experiments at 37 C or provide better justification for why accelerated release was the correct choice.

We thank the reviewer for their comment. While typical dissolution experiments are performed under more physiologically representative conditions (i.e. 37°C), we found running our dissolution at 50°C was more appropriate for the range of coatings we produce. Throughout development and qualification of our in vitro dissolution assay we have consistently found temperature has the biggest impact on the rate of dissolution of ALD-coated products. In an experiment which compared the dissolution behavior of a single 100 cycle ALD-coated powder run at 4°C, 37°C, 45°C, and 50°C, a linear correlation was found between the temperature and the time to release (measured as the time to the inflection point or the reported “C” parameter). Perhaps unsurprisingly, there was an approximately 2.3-fold reduction in release rate when the assay was performed at 50°C vs 37°C. Consequently, if the 500- and 1000- cycle powders shown in this manuscript were run at 37°C, it could take several weeks to capture their full dissolution profile. This extended experimental time increases the possibility of evaporation, microbial contamination, and unnecessary variability in our in vitro dissolution assay. Therefore, we set all our dissolution experiments at 50°C as it provides us adequate discrimination between different cycle numbers, facilitates the full dissolution of our highest cycle number products within a week (as show in Fig 1B), and generates reproducible results. A sentence justifying the accelerated conditions has been added in the dissolution method section.

  • There are no characterization methods used to show what the coated particles look like. At a minimum, they should use scanning electron microscopy images or something similar to show particles before and after coating. Something like this may be included in reference #9 but this paper should be self-contained and not dependent on another paper (that uses a different antigen as well).

Thank you for your comment on particle characterization. We have included additional background in the introduction that describes SEM and TEM methods used to characterize the particles previously reference #9. Similar characterization was conducted on the ovalbumin-containing particles used in this study. No significant differences were observed in comparison to the prior study. Therefore we did not deem this data critical and/or informative for the current study. In the current study, we focus on the new technique (in vitro dissolution) for improved particle coat characterization that is quantitative and easier to perform.

  • There are many places where a single data point is reported, lacking standard deviations and/or error bars (e.g., all of Fig 1 except for 1D, Figure 2A,D, Supp Tables 1 & 2). This needs to be corrected.

We thank the reviewer for their comment on data points and standard deviations and/or error bars. 

For Figure 1 - We added 95% confidence intervals for the anti-OVA total IgG titers reported in Figure 1C. For Figures 1E,1F,1G - these graphs report on pooled serum samples and therefore standard deviations and/or error bars are not suitable to report for this data. The language in the figure legend has been updated to improve clarity on pooling of serum samples. The data from this experiment are not used to make comparative distinctions between the groups (ALD 50 vs Alhydrogel) so statistics are not needed. The data were helpful in establishing that IgG1 titer assessment is suitable for characterizations in Figure 2.

For Figure 2 - We added 95% confidence intervals to the GMT data displayed in Figure 2A. Error bars are not warranted for Figure 2D because the data depict the percentage of the population that has seroconverted.

For Supplementary Tables - We added 95% confidence intervals for the 4PL terms described in the supplementary tables.

  • The authors should discuss that the favorable results obtained in the paper are due to the powder acting like nano alum, which typically outperforms Alhydrogel as an adjuvant.

We thank the reviewer for commenting on the favorable results that we obtained. There is a key difference between ALD coated powders and nano alum. Specifically, antigens are adsorbed to nano alum whereas the antigen is contained within ALD coated particles. This paper is narrowly focused on how ALD coating impacts vaccine kinetics for a delayed-release technology (both in vitro release and in vivo antibody response times). We specifically avoid making claims regarding performance as compared with Alhydrogel, because the goal is to establish a correlation between ALD cycle/coat thickness and vaccine kinetics which can be used in future vaccine designs. Alhydrogel is used in the current study as a control for establishing antibody responses. Comparisons to other adjuvants like Alhydrogel and nano alum are out of the scope for the current paper. However, we added discussion related to this comment to explore as a future research direction. Thank you for the suggestion.

  • The evidence for antibody class switching is very weak, consisting only of IgM and IgG titers in which the IgM titers exceed IgG1 titers by more than 100-fold. It is also unclear if the alum and experimental group are statistically significantly different since there are no error bars or indications of statistical significance. Even if it were, the data should be taken in context since IgM titers are also higher for the ALD 50 group. I would recommend excluding the comment on like 155. Similarly, I do not see evidence in Figure 1 for the response beginning earlier. The ALD 50 and alum groups both appear to respond immediately. If the authors want to keep the corresponding comment on lines 156 and 157, they need to support the claim with data.

We thank thank the reviewer for their comments on Figures 1E,1F, and 1G on specific antibody subtypes. Class switching is defined by a change from IgM to IgA, IgG, or IgE. Thus, we maintain that IgG1 titers are evidence of class switching. The graphs depict analysis of pooled sera, so there is no statistical analysis to report. The high level of OVA-specific IgM is present one week before vaccination (-1 in graph), and may be attributed to a high amount of natural cross-reactivity. We interpret the result as no observable increase in IgM production at the times tested following vaccination. It is also possible that we missed a spike in IgM production (for example at 1 week following vaccination) as these early antibody response can occur and resolve quickly. The titer numbers themselves should not be compared between isotypes, because they are linked to the reported concentration from different commercially available isotype standards (Chondrex). ----- Ultimately, we agree with the reviewers assessment that ALD 50 cycle and Alhydrogel conditions respond with similar kinetics ("immediately")! We added some language to the section being referenced to help clarify this point.

  • The extensive discussion of seroconversion does not make sense and should be removed since there is no key level that is known to be protective from OVA since it is not infectious. Therefore, all these sub-figures and descriptions do is binarize quantitative data without adding value.

Thank you for your comments on seroconversion. We agree with the reviewer that seroconversion is an odd measure for a model antigen like OVA that provides no protection from an infectious agent. Usually, the threshold for seroconversion is established by the antibody titer that provides protection against the infectious agent. However, we maintain that seroconversion is important for our kinetic characterization of ALD coated vaccine products. Specifically, peak titers may not represent the earliest antibody response (as seen by comparison between Figure 2C and 2E). Thus creating a threshold (like seroconversion) that establishes a binary early response can help to characterize when the vaccine starts to have an effect. We think that this is an important distinction. If the goal is to establish an in vitro to in vivo correlation that is predictive of how and when the product works, these parameters need to be considered. Is the goal to establish when a vaccine will provide peak titer or to better understand the pharmacokinetics of the vaccine product? Each analysis may be valuable. Ultimately, we find value in understanding the earliest immune responses as it can be used to establish when the coated products start releasing vaccine product in vivo. Therefore, platform analysis by seroconversion is a useful metric for establishing product characteristics in the context of delayed-release vaccine technologies.

Minor Issues:

  • The authors should comment on the suitability of this method for encapsulating vaccine antigens rather than a model protein. For example, it seems like the fabrication conditions (50 C) would/could change the conformation of the antigen and thus its ability to confer immunity.

Thank you for your comment. This has been demonstrated with HPV L1 capsomers previously in reference #10 (Witeof et al). We will add a line in the discussion highlighting this important thermostability feature of making vaccines with spray drying and atomic layer deposition.

  • The authors should discuss what a reasonable number of cycles would be for a commercial product. For example, would 1,000 cycles be commercially viable or too expensive?

We thank the reviewer for their comment. We do not think that 1,000 cycle products would be a good fit for clinical use at this time. Importantly, there are limits to the amount of aluminum that can be included in human vaccine products. At some point these limits will be reached by iterative ALD coating. But more importantly, delaying the release of vaccine beyond 30 weeks is not necessary. Creating combination prime and boost products with shorter delays will provide sufficient immunity, be more cost effective, and conform to the limits of aluminum content for human vaccines.

  • The legend in Figure 1B is missing the color markings. I believe two asterisks are also missing from the comparator bar in Figures 1D and 2B.

We thank the reviewer for their comment. It isn't clear why the color markings and asterisk symbols did not appear in the pdf version that you received. We are attempting to save the figure files in a different format to address this issue. We will also be sure to confirm that these are present in the final publication proofs.

  • Can the authors please explain how they are getting antibodies in ng/mL? I am not sure how this number can be obtained without a number of assumptions and it is not standard to report this type of unit.

The antibody titers are calculated with a standard curve that is commercially available (BioLegend and Chondrex). These standards are predefined by the manufacturer with specifications that include a concentration. For example, IgG1 monoclonal antibody against ovalbumin is provided at 1mg/mL (Chondrex Cat# 7093). Dilutions of this antibody are used to build a standard curve and the unknown titer concentrations from serum samples are interpolated from the standard curve.

Reviewer 2 Report

Comments and Suggestions for Authors

Dear authors, 
I found your work interesting and with potential for further studies. However, I have some remarks. 
1) Section 2.1 - You could mention how many coating cycles have been performed
2) line 97 - different fonts were used
3) line 104 - which version of Prism software were used for statistical analysis?
4) Section 3. Results - in this section you have reported results and discuss them and it would be better if titled "Results and discussion"
5) Section 4 - sounds more as conclusion than as discussion. You might title it "Conclusions" or merge it with previous section.

Author Response

Dear Reviewer 2,

Thank you for your comments on our manuscript, we found them to be helpful and are happy to incorporate changes to clarify and/or address them specifically.

1) Section 2.1 - You could mention how many coating cycles have been performed

Thank you for this comment. We agree that including the number of ALD coating cycles in the materials and methods section 2.1 will improve clarity for our manuscript. Therefore, we have added a sentence to the end of the first paragraph in section 2.1. The new sentence reads as follows: "The number of ALD cycles used to coat powders in this study were 50, 100, 250, 500, or 1000."

2) line 97 - different fonts were used

Thank you for pointing this out. The different fonts have been corrected. 

3) line 104 - which version of Prism software were used for statistical analysis?

Thank you for pointing this out. The statistical analysis section has been updated to include the version of Prism software used in this report "version 10.2.3".

4) Section 3. Results - in this section you have reported results and discuss them and it would be better if titled "Results and discussion"

Thank you for this comment.  The discussion section has been expanded. We hope that you find this helpful.

5) Section 4 - sounds more as conclusion than as discussion. You might title it "Conclusions" or merge it with previous section.

Thank you for this comment. A separate "Conclusions" has been added. We hope that you find this helpful.

Reviewer 3 Report

Comments and Suggestions for Authors

This manuscript introduces the generation of a delayed-release vaccine using atomic layer deposition. The results show that increasing the thickness of the coating would affect the kinetics of antigen release and antibody response. Some minor issues need to be addressed as follows.

1. In the title of this paper, it should be "delayed-release vaccine".

2. Figure 1C, 1E, 1F and 1G should have statistical parameters, e.g. error bars.

3. In Figure 1D and 2B, the top lines are missing the significance level.

4. The supplementary table 1 and 2 should also include statistical parameters.

Author Response

Dear Reviewer 3, Thank you for your comments. We found them to be helpful and provide the following comments and/or edits to address them accordingly.

1. In the title of this paper, it should be "delayed-release vaccine".

Thank you for this comment. We have updated the title and and the abstract to include the hyphen with using the term "delayed-release vaccine".

2. Figure 1C, 1E, 1F and 1G should have statistical parameters, e.g. error bars.

Thank you for your question on the statistical parameters for Figures 1C, 1E, 1F, and 1G. Figure 1C has been updated to include the 95% confidence intervals from the geometric mean titer calculated on five mouse serum samples. Figures 1E, 1F, and 1G do not have any statistical parameters to report. The data reported for 1E, 1F, and 1G are from pooled sera (five samples combined at equal volume), thus it represents an average antibody response for each experimental condition. The legend for Figure 1 has been updated to improve clarity.

3. In Figure 1D and 2B, the top lines are missing the significance level.

Thank you for pointing this out. The significance levels did not appear in the .pdf version of the manuscript. However, the asterisks can be seen in the figure .pdf files and in the Word .docx file. It isn't clear why this happened. However, we will be sure that they are included in the final proofs.

4. The supplementary table 1 and 2 should also include statistical parameters.

Thank you for this comment. We have updated supplementary table 1 and 2 to include the 95% confidence intervals for each term (when they could be calculated). The legends for each table were also updated accordingly.